# Cumulative Residual *q*-Fisher Information and Jensen-Cumulative Residual *χ*^2^ Divergence Measures

**DOI:** 10.3390/e24030341

**Published:** 2022-02-27

**Authors:** Omid Kharazmi, Narayanaswamy Balakrishnan, Hassan Jamali

**Affiliations:** 1Department of Statistics, Faculty of Mathematical Sciences, Vali-e-Asr University of Rafsanjan, Rafsanjan P.O. Box 518, Iran; 2Department of Mathematics and Statistics, McMaster University, Hamilton, ON L8S 4L8, Canada; bala@mcmaster.ca; 3Department of Mathematics, Faculty of Mathematical Sciences, Vali-e-Asr University of Rafsanjan, Rafsanjan P.O. Box 518, Iran; jamali@vru.ac.ir

**Keywords:** Shannon entropy, *q*-Fisher information, cumulative residual Fisher information, Jensen inequality, hazard function, cumulative residual chi-square divergence, *q*-hazard rate function

## Abstract

In this work, we define cumulative residual *q*-Fisher (CRQF) information measures for the survival function (SF) of the underlying random variables as well as for the model parameter. We also propose *q*-hazard rate (QHR) function via *q*-logarithmic function as a new extension of hazard rate function. We show that CRQF information measure can be expressed in terms of the QHR function. We define further generalized cumulative residual $\chi^{2}$ divergence measures between two SFs. We then examine the cumulative residual *q*-Fisher information for two well-known mixture models, and the corresponding results reveal some interesting connections between the cumulative residual *q*-Fisher information and the generalized cumulative residual $\chi^{2}$ divergence measures. Further, we define Jensen-cumulative residual $\chi^{2}$ (JCR-$\chi^{2}$) measure and a parametric version of the Jensen-cumulative residual Fisher information measure and then discuss their properties and inter-connections. Finally, for illustrative purposes, we examine a real example of image processing and provide some numerical results in terms of the CRQF information measure.

## 1. Introduction

Entropy-type and Fisher-type information measures have attracted great attention from researchers in information theory. Although these two types of information measures have different geneses, they are complementary to each other in the study of information resources. Among them, Fisher information and Shannon entropy are the fundamental information measures and have been used very broadly. The systems with complex structure can be thoroughly described in terms of their architecture (Fisher information) and their behavior (Shannon entropy) measures. For more details, one may refer to Cover and Thomas [1] and Zegers [2]. Fisher [3] had proposed an information measure for describing the interior properties of a probabilistic model. Shannon entropy originated from the pioneering work of Shannon [4], based on a study of the global behavior of systems modeled by a probability structure. Fisher information as well as Shannon entropy are quite important and have become fundamental quantities in numerous applied disciplines. These information measures and their extensions have been considered by several researchers in recent years. In addition, some divergence measures, based on Fisher information and Shannon entropy, have been introduced for measuring similarity and dissimilarity between two statistical models. For example, Kullback-Leibler, chi-square, and relative Fisher information divergences and their extensions have been used in this regard. For pertinent details, one may refer to Nielsen and Nock [5], Zegers [2], Popescu et al. [6], Sánchez-Moreno et al. [7], and Bercher [8].

Tsallis entropy (see [9]) is a generalized form of Shannon entropy and has found key applications in the context of non-extensive thermo-statistics. Some generalized forms of Fisher information that match well in the context of non-extensive thermo-statistics have also been introduced by some authors; see Johnson and Vignat [10], Furuichi [11], and Lutwak et al. [12]. One of the common ways of generalizing information measures is through their accumulated forms. This has been done for Shannon entropy, Tsallis entropy, and their related versions.

Let *X* denote a continuous random variable on the support X with survival function ${\bar{F}}_{\theta }\left(x\right)$. The cumulative residual Fisher (CRF) information measure, introduced by Kharazmi and Balakrishnan [13], is then defined as
(1)$$\begin{array}{ccc} {\mathrm{CI}}_{\theta }\left({\bar{F}}_{\theta }\right) & = & \int_{X}{\bar{F}}_{\theta }\left(x\right){\left(\frac{\partial \log{\bar{F}}_{\theta }\left(x\right)}{\partial \theta }\right)}^{2}dx, \end{array}$$
where log stands for the natural logarithm. Throughout this paper, we will suppress X in the integration with respect to *x* for ease of notation, unless a distinction becomes necessary. Due to the term ${\bar{F}}_{\theta }$ in the integrand in (Equation 1), it can be readily seen that CRF information measure in (Equation 1) provides decreasing weights for larger values of *X*. Hence, this information measure will naturally be robust to the presence of outliers. Kharazmi and Balakrishnan [13] have analogously defined the CRF information measure for the survival function $\left(\bar{F}\right)$ as
(2)$$\begin{array}{c} \mathrm{CI}\left(X\right)=\mathrm{CI}\left(\bar{F}\right)=\int \bar{F}\left(x\right){\left\{\frac{\partial \log\bar{F}\left(x\right)}{\partial x}\right\}}^{2}dx. \end{array}$$

They also provided an interesting representation for CI(X) information measure in (Equation 2), based on the hazard function, $r_{F}\left(x\right)=f(x)/\bar{F}\left(x\right)$, as
(3)$$\begin{array}{ccc} \mathrm{CI}(X) & = & E[r_{F}\left(X\right)]. \end{array}$$

In the present paper, our primary goal is to introduce cumulative residual *q*-Fisher (CRQF) information, cumulative residual generalized-$\chi^{2}$ (CRG-$\chi^{2}$) divergence, and Jensen-cumulative residual-$\chi^{2}$ (JCR-$\chi^{2}$) and a parametric version of Jensen-cumulative residual Fisher (JCRF) information measures. We then examine some properties of these information measures and their interconnections in terms of two well-known mixture models that are commonly used in reliability, economics, and survival analysis.

The organization of the rest of this paper is as follows. In Section 2, we briefly describe some key information and entropy measures that are essential for all subsequent developments. Next, in Section 3, we define a cumulative residual *q*-Fisher (CRQF) information measure and *q*-hazard rate function. It is then shown that the CRQF information measure can be expressed via expectation involving *q*-hazard rate (QHR) function under proportional hazard model (PH) with proportionality parameter *q*. In Section 4, we propose the cumulative version of a generalized chi-square measure, called cumulative residual generalized-$\chi^{2}$ (CRG-$\chi^{2}$) divergence measure. We show that the first derivative of CRG-$\chi^{2}$ measure with respect to the associated parameter is connected to the cumulative residual entropy measure, and when the parameter tends to zero, it is connected to the variance of the ratio of two survival functions. Next, we obtain the CRQF information measure for two well-known mixture models in Section 5. It is shown that the CRQF information measure for arithmetic mixture and harmonic mixture models are connected to the CRG-$\chi^{2}$ divergence measure. In addition, we show that the harmonic mixture model involves optimal information under three optimization problems regarding the cumulative residual chi-square divergence measure. In Section 6, we first define Jensen-cumulative residual $\chi^{2}$ (JCR-$\chi^{2}$) measure and a parametric version of the Jensen-cumulative residual Fisher information measure and then discuss some of their properties. We also show that these two information measures are connected through arithmetic mixture models. In Section 7, we consider a real example of image processing and present some numerical results in terms of the CRQF information measure. Finally, some concluding remarks are made in Section 8.

## 2. Preliminaries

In this section, we briefly review some information measures that will be used in the sequel. The chi-square divergence between two SFs $\bar{F}$ and $\bar{G}$, called cumulative residual $\chi^{2}$ (CR-$\chi^{2}$) divergence, is defined as
(4)$$\begin{array}{c} \chi^{2}\left(\bar{F},\bar{G}\right)=\int \frac{{\left(\bar{G}\left(x\right)-\bar{F}\left(x\right)\right)}^{2}}{\bar{F}\left(x\right)}dx. \end{array}$$

The $\chi^{2}\left(\bar{G},\bar{F}\right)$ divergence can also be defined in an analogous manner. For more details, see Kharazmi and Balakrishnan [13].

For a given continuous random variable *X* with survival function $\bar{F}\left(x\right)$, the cumulative residual entropy (CRE) was defined by Rao et al. [14] as
(5)$$\xi \left(X\right)=\int \left\{-\log\bar{F}\left(x\right)\right\}\bar{F}\left(x\right)dx.$$

The relative cumulative residual Fisher (RCRF) information between two absolutely continuous survival functions ${\bar{G}}_{\theta }$ and ${\bar{F}}_{\theta }$ is defined as
(6)$$\begin{array}{c} \mathrm{CD}\left({\bar{F}}_{\theta },{\bar{G}}_{\theta }\right)=\int {\left\{\frac{\partial \log{\bar{G}}_{\theta }\left(x\right)}{\partial \theta }-\frac{\partial \log{\bar{F}}_{\theta }\left(x\right)}{\partial \theta }\right\}}^{2}{\bar{F}}_{\theta }\left(x\right)dx. \end{array}$$

For given absolutely continuous survival functions ${\bar{F}}_{1},\cdots ,{\bar{F}}_{n}$, the Jensen-cumulative residual Fisher (JCRF) information measure was defined by Kharazmi and Balakrishnan [13] as
(7)$$\begin{array}{ccc} \mathrm{JCI}({\bar{F}}_{1},\cdots ,{\bar{F}}_{n};\alpha ) & = & \sum_{i=1}^{n}\alpha_{i}\mathrm{CI}\left({\bar{F}}_{i}\right)-\mathrm{CI}\left(\sum_{i=1}^{n}\alpha_{i}{\bar{F}}_{i}\right), \end{array}$$
where $\alpha_{1},\cdots ,\alpha_{n}$ are non-negative real values with $\sum_{i=1}^{n}\alpha_{i}=1$.

## 3. CRQF Information Measure

Here, we first define the cumulative residual *q*-Fisher (CRQF) information measure and the *q*-hazard rate (QHR) function. We then study some properties of the CRQF information measure and its connection to the QHR function.

The *q*-Fisher information of a density function *f*, defined by Lutwak et al. [12], is given by
(8)$$\begin{array}{c} I_{q}\left(f\right)=\int {\left(\frac{\partial \log_{q}f\left(x\right)}{\partial x}\right)}^{2}f\left(x\right)dx, \end{array}$$
where $\log_{q}\left(x\right)$ is the *q*-logarithmic function defined as
(9)$$\begin{array}{c} \log_{q}\left(x\right)=\frac{x^{q}-1}{q},\;\;x\in ℜ,\;q\neq 0. \end{array}$$

For more details, see Furuichi [11], Yamano [15], and Masi [16]. Using this *q*-logarithmic function, we now propose two cumulative versions of the *q*-Fisher information in (Equation 8).

**Definition** **1.**Let ${\bar{F}}_{\theta }$ denote the survival function of variable *X*. Then, the CRQF information about parameter θ is defined as
(10)$$\begin{array}{ccc} {\mathrm{CI}}_{q}\left(\theta \right) & = & \int {\left(\frac{\partial \log_{q}{\bar{F}}_{\theta }\left(x\right)}{\partial \theta }\right)}^{2}{\bar{F}}_{\theta }\left(x\right)dx. \end{array}$$

**Definition** **2.**The CRQF information measure for survival function $\bar{F}$ associated with variable *X* is defined as
(11)$$\begin{array}{c} {\mathrm{CI}}_{q}\left(X\right)={\mathrm{CI}}_{q}\left(\bar{F}\right)=\int {\left\{\frac{\partial \log_{q}\bar{F}\left(x\right)}{\partial x}\right\}}^{2}\bar{F}\left(x\right)dx. \end{array}$$

**Example** **1.**
*Let X have a Weibull distribution with CDF $F\left(x\right)=1-e^{-\lambda x^{\beta }}$, x,β,λ>0. Then, the CRQF information measure of variable X, for $\beta >\frac{1}{2}$, is obtained as*

(12)
$$\begin{array}{c} {\mathrm{CI}}_{q}\left(X\right)=\frac{\beta \lambda^{\frac{1}{\beta }}\Gamma \left(2-\frac{1}{\beta }\right)}{{\left(1+2q\right)}^{2-\frac{1}{\beta }}}, \end{array}$$

*where Γ(.) is the complete gamma function defined by $\Gamma \left(a\right)=\int_{0}^{\infty }x^{a-1}e^{-x}dx.$*


Figure 1 plots the CRQF information measure in (Equation 12) for different choices of parameters, from which we observe that ${\mathrm{CI}}_{q}\left(X\right)$ gets maximized when *q* is decreased and the parameter λ is increased.

### q-Hazard Rate Function and Its Connection to CRQF Information Measure

The hazard rate (HR) function of variable *X* with survival function $\bar{F}$ is given by
(13)$$\begin{array}{c} r\left(x\right)=\frac{\partial \log\bar{F}\left(x\right)}{\partial x}=\frac{f(x)}{\bar{F}\left(x\right)}. \end{array}$$

The HR function is a basic concept in reliability theory; see Barlow and Proschan [17] for elaborate details.

Now, we first propose a new extension of the HR function based on the *q*-logarithmic function in (Equation 9) and then study its connection to the CRQF information measure.

**Definition** **3.**For a random variable *X* with an absolutely continuous survival function $\bar{F}$, the *q*-hazard rate (QHR) (or *q*-logarithmic hazard rate) function is defined as
(14)$$r_{q}\left(x\right)=\frac{\partial \log_{q}\bar{F}\left(x\right)}{\partial x}=\left\{\begin{array}{cc} r\left(x\right){\bar{F}}^{q}\left(x\right), & q\neq 0,\\ r(x), & q=0, \end{array}\right.$$
where r(x) is the hazard rate function defined in (Equation 13).

**Theorem** **1.**
*Let variable X have its absolutely continuous survival function $\bar{F}$ and q-hazard rate function $r_{q}\left(x\right)$ for x>0. Then, for q>0, we have*

$${\mathrm{CI}}_{q}\left(\bar{F}\right)=\frac{1}{q+1}E_{X_{q}}\left[r_{q}\left(X\right)\right],$$

*where $X_{q}$ has proportional hazard model corresponding to baseline variable X with proportionality parameter q.*


**Proof.** From the definition of ${\mathrm{CI}}_{q}\left(\bar{F}\right)$ in (Equation 11), we get
$$\begin{array}{ccc} {\mathrm{CI}}_{q}\left(\bar{F}\right) & = & \int_{0}^{\infty }{\left\{\frac{\partial \log_{q}\bar{F}\left(x\right)}{\partial x}\right\}}^{2}\bar{F}\left(x\right)dx\\ & = & \int_{0}^{\infty }r_{q}\left(x\right)f\left(x\right){\bar{F}}^{q}\left(x\right)dx\\ & = & \frac{1}{q+1}E_{X_{q}}\left[r_{q}\left(X\right)\right], \end{array}$$
as required. □

**Lemma** **1.**
*The ${\mathrm{CI}}_{q}\left(\bar{F}\right)$ information measure is decreasing with respect to q>0.*


**Proof.** From the definition of ${\mathrm{CI}}_{q}\left(F\right)$ in (Equation 11), upon making use of Theorem 1, for each $0<q_{1}\leq q_{2}$, we have
$$\begin{array}{ccc} {\mathrm{CI}}_{q_{1}}\left(\bar{F}\right) & = & \int_{0}^{\infty }{r_{q}}_{1}\left(x\right)f\left(x\right){\bar{F}}^{q_{1}}\left(x\right)dx\\ & = & \int_{0}^{\infty }r\left(x\right)f\left(x\right){\bar{F}}^{2q_{1}}\left(x\right)dx\\ & \geq & \int_{0}^{\infty }r\left(x\right)f\left(x\right){\bar{F}}^{2q_{2}}\left(x\right)dx\\ & = & \int_{0}^{\infty }r_{q_{2}}\left(x\right)f\left(x\right){\bar{F}}^{q_{2}}\left(x\right)dx\\ & = & {{\mathrm{CI}}_{q}}_{2}\left(\bar{F}\right), \end{array}$$
as required. □

**Theorem** **2.**
*Let the non-negative random variable X have survival function $\bar{F}$, CRF information $\mathrm{CI}(\bar{F})$, and CRQF information ${\mathrm{CI}}_{q}\left(\bar{F}\right)$. Then, we have:*
 *(i)* 
*If q>0, then ${\mathrm{CI}}_{q}\left(\bar{F}\right)\leq \mathrm{CI}\left(\bar{F}\right)$;*
 *(ii)* 
*If q<0, then ${\mathrm{CI}}_{q}\left(\bar{F}\right)\geq \mathrm{CI}\left(\bar{F}\right)$’*

*with equality holding if and only if q=0.*


**Proof.** From the definition of CRQF information measure in (Equation 11), and since ${\bar{F}}^{q}\left(x\right)\leq 1$ for q>0, we have
$$\begin{array}{ccc} {\mathrm{CI}}_{q}\left(\bar{F}\right) & = & \int_{0}^{\infty }{\left\{\frac{\partial \log_{q}\bar{F}\left(x\right)}{\partial x}\right\}}^{2}\bar{F}\left(x\right)dx\\ & = & \int_{0}^{\infty }r_{q}\left(x\right)f\left(x\right){\bar{F}}^{q}\left(x\right)dx\\ & = & \int_{0}^{\infty }r\left(x\right)f\left(x\right){\bar{F}}^{2q}\left(x\right)dx\\ & \leq & \int_{0}^{\infty }r\left(x\right)f\left(x\right)dx\\ & = & \mathrm{CI}(\bar{F}), \end{array}$$
which proves Part (i). Part (ii) can be proved in a similar manner. □

## 4. Generalized Cumulative Residual $\chi^{2}$ Divergence Measures

In this section, we first define a cumulative form of a generalized chi-square divergence measure and then examine some of its properties. A generalized version of the $\chi^{2}$ divergence between two densities *f* and *g*, for α≥0, considered by Basu et al. [18], is defined as
(15)$$\begin{array}{c} \chi_{\alpha }^{2}\left(f,g\right)=\frac{1+\alpha }{2}\int \frac{{\left(g(x)-f(x)\right)}^{2}}{f^{1-\alpha }\left(x\right)}dx. \end{array}$$

For more details, see also Ghosh et al. [19].

**Definition** **4.**
*The cumulative residual generalized $\chi^{2}$ (CRG-$\chi^{2}$) divergence between two survival functions $\bar{F}$ and $\bar{G}$, for α≥0, is defined as*

(16)
$$\begin{array}{c} \chi_{\alpha }^{2}\left(\bar{F},\bar{G}\right)=\frac{\alpha +1}{2}\int \frac{{\left(\bar{F}\left(x\right)-\bar{G}\left(x\right)\right)}^{2}}{{\bar{F}}^{1-\alpha }\left(x\right)}dx. \end{array}$$



It is readily seen from (Equation 16) that
$$\chi^{2}\left(\bar{F},\bar{G}\right)=\lim_{\alpha ⟶0^{+}}\frac{\chi_{\alpha }^{2}\left(\bar{F},\bar{G}\right)}{2},$$
and we also have
$$\chi_{\alpha }^{2}\left(\bar{F},\bar{G}\right)\leq \frac{\alpha +1}{2}\chi^{2}\left(\bar{F},\bar{G}\right).$$

**Theorem** **3.**
*Let the variables X and Y have survival functions $\bar{F}$ and $\bar{G}$, respectively, and $\chi_{\alpha }^{2}\left(\bar{F},\bar{G}\right)$ be the corresponding CRG-$\chi^{2}$ divergence measure between them. Then, we have*

$$\begin{array}{ccc} \frac{\partial }{\partial \alpha }\chi_{\alpha }^{2}\left(\bar{F},\bar{G}\right){|}_{\alpha =0} & = & \frac{1}{2}\left\{\chi^{2}\left(\bar{F},\bar{G}\right)-\xi \left(\bar{F}\right)+2K\left(\bar{G},\bar{F}\right)+\int \frac{{\bar{G}}^{2}\left(x\right)}{\bar{F}\left(x\right)}\log\bar{F}\left(x\right)dx\right\}, \end{array}$$

*where $\xi (\bar{F})$ is the CRE information measure defined in (Equation 5) and $K(\bar{G},\bar{F})$ is the cumulative residual inaccuracy measure given by*

$$K\left(\bar{G},\bar{F}\right)=-\int \bar{G}\left(x\right)\log\bar{F}\left(x\right)dx.$$



**Proof.** Upon considering the CRG-$\chi^{2}$ measure in (Equation 16) and differentiating it with respect to α, we obtain $$\begin{array}{ccc} \frac{\partial }{\partial \alpha }\chi_{\alpha }^{2}\left(\bar{F},\bar{G}\right){|}_{\alpha =0} & = & \frac{1}{2}\int \frac{{\left(\bar{G}\left(x\right)-\bar{F}\left(x\right)\right)}^{2}}{\bar{F}\left(x\right)}dx+\frac{1}{2}\int \frac{{\left(\bar{G}\left(x\right)-\bar{F}\left(x\right)\right)}^{2}}{\bar{F}\left(x\right)}\log\bar{F}\left(x\right)dx\\ & = & \frac{1}{2}\chi^{2}\left(\bar{F},\bar{G}\right)+\frac{1}{2}\int \left(\bar{F}\left(x\right)-2\bar{G}\left(x\right)+\frac{{\bar{G}}^{2}\left(x\right)}{\bar{F}\left(x\right)}\right)\log\bar{F}\left(x\right)dx\\ & = & \frac{1}{2}\chi^{2}\left(\bar{F},\bar{G}\right)+\frac{1}{2}\int \bar{F}\left(x\right)\log\bar{F}\left(x\right)dx-\int \bar{G}\left(x\right)\log\bar{F}\left(x\right)dx\\ & & +\frac{1}{2}\int \frac{{\bar{G}}^{2}\left(x\right)}{\bar{F}\left(x\right)}\log\bar{F}\left(x\right)dx\\ & = & \frac{1}{2}\left\{\chi^{2}\left(\bar{F},\bar{G}\right)-\xi \left(\bar{F}\right)+2K\left(\bar{G},\bar{F}\right)+\int \frac{{\bar{G}}^{2}\left(x\right)}{\bar{F}\left(x\right)}\log\bar{F}\left(x\right)dx\right\}, \end{array}$$ as required. □

**Theorem** **4.**
*Let the non-negative continuous variables X and Y have survival functions $\bar{F}$ and $\bar{G}$, respectively, and have a common mean μ. Then,*

(17)
$$\begin{array}{c} \lim_{\alpha ⟶0^{+}}\chi_{\alpha }^{2}\left(\bar{G},\bar{F}\right)=\frac{\mu }{2}Var_{f_{e}}\left(T(X)\right), \end{array}$$

*where $T\left(x\right)=\frac{\bar{G}\left(x\right)}{\bar{F}\left(x\right)}$ and $f_{e}\left(x\right)=\frac{\bar{F}\left(x\right)}{\mu }$ is the equilibrium distribution of variable X.*


**Proof.** From the definition of GCR-$\chi^{2}$ divergence measure in (Equation 16) and the facts that $\int \bar{F}\left(x\right)dx=\int \bar{G}\left(x\right)dx=\mu$, we obtain
$$\begin{array}{ccc} 2\lim_{\alpha ⟶0^{+}}\chi_{\alpha }^{2}\left(\bar{F},\bar{G}\right) & = & \int \frac{{\left(\bar{G}\left(x\right)-\bar{F}\left(x\right)\right)}^{2}}{\bar{F}\left(x\right)}dx\\ & = & \int \frac{{\bar{G}}^{2}\left(x\right)}{\bar{F}\left(x\right)}dx+\int \bar{F}\left(x\right)dx-2\int \bar{G}\left(x\right)dx\\ & = & \mu \int {\left(\frac{\bar{G}\left(x\right)}{\bar{F}\left(x\right)}\right)}^{2}\frac{\bar{F}\left(x\right)}{\mu }dx-\mu {\left(\int \frac{\bar{G}\left(x\right)}{\bar{F}\left(x\right)}\frac{\bar{F}\left(x\right)}{\mu }dx\right)}^{2}\\ & = & \mu \int {\left(\frac{\bar{G}\left(x\right)}{\bar{F}\left(x\right)}\right)}^{2}f_{e}\left(x\right)dx-\mu {\left(\int \frac{\bar{G}\left(x\right)}{\bar{F}\left(x\right)}f_{e}\left(x\right)dx\right)}^{2}\\ & = & \mu Var_{f_{e}}\left(\frac{\bar{G}\left(X\right)}{\bar{F}\left(X\right)}\right)\\ & = & \mu Var_{f_{e}}\left(T(X)\right), \end{array}$$
as required. □

**Theorem** **5.**
*Let the variables X and Y have survival functions $\bar{F}$ and $\bar{G}$, respectively, and $\chi_{\alpha }^{2}\left(\bar{F},\bar{G}\right)$ be the corresponding CRG-$\chi^{2}$ divergence measure between them. Then, for each $0\leq \alpha_{1}\leq \alpha_{2}$, we have*

$$\begin{array}{c} \chi_{\alpha_{1}}^{2}\left(\bar{F},\bar{G}\right)\geq \frac{\alpha_{1}+1}{\alpha_{2}+1}\chi_{\alpha_{2}}^{2}\left(\bar{F},\bar{G}\right). \end{array}$$



**Proof.** From the definition of CRG-$\chi^{2}$ in (Equation 16) and that fact that ${\bar{F}}^{\alpha_{1}}\left(x\right)\geq {\bar{F}}^{\alpha_{1}}\left(x\right)$ for $0\leq \alpha_{1}\leq \alpha_{2}$, we have
$$\begin{array}{ccc} \frac{2}{\alpha_{1}+1}\chi_{\alpha_{1}}^{2}\left(\bar{F},\bar{G}\right) & = & \int \frac{{\left(\bar{G}\left(x\right)-\bar{F}\left(x\right)\right)}^{2}}{{\bar{F}}^{1-\alpha_{1}}\left(x\right)}dx\\ & \geq & \int \frac{{\left(\bar{G}\left(x\right)-\bar{F}\left(x\right)\right)}^{2}}{{\bar{F}}^{1-\alpha_{2}}\left(x\right)}dx\\ & = & \frac{2}{\alpha_{2}+1}\chi_{\alpha_{2}}^{2}\left(\bar{F},\bar{G}\right), \end{array}$$
as required. □

## 5. Cumulative Residual q-Fisher Information for Two Well-Known Mixture Models

In this section, we study the CRQF information measure for the well-known arithmetic mixture and harmonic mixture distributions.

### 5.1. Arithmetic Mixture Distribution

The arithmetic mixture distribution based on survival functions ${\bar{F}}_{1}$ and ${\bar{F}}_{2}$ is given by
(18)$$\begin{array}{c} {\bar{F}}_{\eta }\left(x\right)=\eta {\bar{F}}_{1}\left(x\right)+\left(1-\eta \right){\bar{F}}_{2}\left(x\right),\;\eta \in \left(0,1\right). \end{array}$$

For more details about the mixture model in (Equation 18), see Marshall and Olkin [20]. The CRQF information measure about parameter η in (Equation 18) is given by
(19)$$\begin{array}{ccc} {\mathrm{CI}}_{q}\left(\eta \right) & = & \int {\left(\frac{\partial \log_{q}{\bar{F}}_{\eta }\left(x\right)}{\partial \eta }\right)}^{2}{\bar{F}}_{\eta }\left(x\right)dx\\ & = & \int \frac{{\left({\bar{F}}_{2}\left(x\right)-{\bar{F}}_{1}\left(x\right)\right)}^{2}}{{\bar{F}}_{\eta }^{1-2q}\left(x\right)}dx. \end{array}$$

**Theorem** **6.**
*The CRQF information measure in (Equation 19) is given by*

$$\begin{array}{ccc} {\mathrm{CI}}_{q}\left(\eta \right) & = & \frac{8}{2q+1}M_{\frac{1}{2}}\left(\chi_{2q}^{2}\left({\bar{F}}_{\eta },{\bar{F}}_{1}\right),\chi_{2q}^{2}\left({\bar{F}}_{\eta },{\bar{F}}_{2}\right)\right), \end{array}$$

*where $M_{\frac{1}{2}}\left(.,.\right)$ is power mean with exponent $\frac{1}{2}$, defined as $M_{\frac{1}{2}}\left(x,y\right)={\left(\frac{x^{\frac{1}{2}}}{2}+\frac{y^{\frac{1}{2}}}{2}\right)}^{2}$ for positive x and y.*


**Proof.** From the mixture model in (Equation 18), we have
(20)$$\begin{array}{c} {\bar{F}}_{2}\left(x\right)-{\bar{F}}_{1}\left(x\right)=\frac{{\bar{F}}_{\eta }\left(x\right)-{\bar{F}}_{1}\left(x\right)}{1-\eta }=\frac{{\bar{F}}_{2}\left(x\right)-{\bar{F}}_{\eta }\left(x\right)}{\eta }. \end{array}$$Now, from the definition of the CQTF information measure in (Equation 19), we find
(21)$${\mathrm{CI}}_{q}\left(\eta \right)=\int \frac{{\left({\bar{F}}_{2}\left(x\right)-{\bar{F}}_{1}\left(x\right)\right)}^{2}}{{\bar{F}}_{\eta }^{1-2q}\left(x\right)}dx=\left\{\begin{array}{cc} \frac{2}{\left(1+2q\right){\left(1-\eta \right)}^{2}}\chi_{2q}^{2}\left({\bar{F}}_{\eta },\bar{F}\right), & \bar{F}={\bar{F}}_{1},\\ \frac{2}{\left(1+2q\right)\eta^{2}}\chi_{2q}^{2}\left({\bar{F}}_{\eta },\bar{F}\right), & \bar{F}={\bar{F}}_{2}. \end{array}\right.$$Then, from (Equation 21), we obtain
$$\begin{array}{ccc} {\mathrm{CI}}_{q}\left(\eta \right) & = & \frac{2}{1+2q}{\left(\sqrt{\chi_{2q}^{2}\left({\bar{F}}_{\eta },{\bar{F}}_{1}\right)}+\sqrt{\chi_{2q}^{2}\left({\bar{F}}_{\eta },{\bar{F}}_{2}\right)}\right)}^{2}\\ & = & \frac{8}{1+2q}{\left(\frac{1}{2}\sqrt{\chi_{2q}^{2}\left({\bar{F}}_{\eta },F_{1}\right)}+\frac{1}{2}\sqrt{\chi_{2q}^{2}\left({\bar{F}}_{\eta },{\bar{F}}_{2}\right)}\right)}^{2}\\ & = & \frac{8}{1+2q}M_{\frac{1}{2}}\left(\chi_{2q}^{2}\left({\bar{F}}_{\eta },{\bar{F}}_{1}\right),\chi_{2q}^{2}\left({\bar{F}}_{\eta },{\bar{F}}_{2}\right)\right), \end{array}$$
as required. □

**Theorem** **7.**
*Let the non-negative random variable X have arithmetic mixture survival function in (Equation 18) and with CRF information CI(η) and CRQF information ${\mathrm{CI}}_{q}\left(\eta \right)$. Then, we have:*
 *(i)* 
*If q>0, then ${\mathrm{CI}}_{q}\left(\eta \right)\leq \mathrm{CI}\left(\eta \right)$;*
 *(ii)* 
*If q<0, then ${\mathrm{CI}}_{q}\left(\eta \right)\geq \mathrm{CI}\left(\eta \right)$,*

*with equality holding if and only if q=0.*


**Proof.** From the definition of CRF information measure, we have
$$\begin{array}{ccc} \mathrm{CI}(\eta ) & = & \int {\left(\frac{\partial \log{\bar{F}}_{\eta }\left(x\right)}{\partial \eta }\right)}^{2}{\bar{F}}_{\eta }\left(x\right)dx\\ & = & \int \frac{{\left({\bar{F}}_{2}\left(x\right)-{\bar{F}}_{1}\left(x\right)\right)}^{2}}{{\bar{F}}_{\eta }\left(x\right)}dx\\ & = & {\left(1-\eta \right)}^{2}\int \frac{{\left({\bar{F}}_{\eta }\left(x\right)-{\bar{F}}_{1}\left(x\right)\right)}^{2}}{{\bar{F}}_{\eta }\left(x\right)}dx\\ & = & {\left(1-\eta \right)}^{2}\chi^{2}\left({\bar{F}}_{\eta },{\bar{F}}_{1}\right). \end{array}$$Furthermore, from the definition of CRTF information measure, we find
$$\begin{array}{ccc} {\mathrm{CI}}_{q}\left(\eta \right) & = & \int {\left(\frac{\partial \log_{q}{\bar{F}}_{\eta }\left(x\right)}{\partial \eta }\right)}^{2}{\bar{F}}_{\eta }\left(x\right)dx\\ & = & \int \frac{{\left({\bar{F}}_{2}\left(x\right)-{\bar{F}}_{1}\left(x\right)\right)}^{2}}{{\bar{F}}_{\eta }\left(x\right)}{\bar{F}}_{\eta }^{2q}\left(x\right)dx\\ & = & {\left(1-\eta \right)}^{2}\int \frac{{\left({\bar{F}}_{\eta }\left(x\right)-{\bar{F}}_{1}\left(x\right)\right)}^{2}}{{\bar{F}}_{\eta }\left(x\right)}{\bar{F}}_{\eta }^{2q}\left(x\right)dx\\ & \overset{q>0}{\leq } & {\left(1-\eta \right)}^{2}\int \frac{{\left({\bar{F}}_{\eta }\left(x\right)-{\bar{F}}_{1}\left(x\right)\right)}^{2}}{{\bar{F}}_{\eta }\left(x\right)}dx\\ & = & {\left(1-\eta \right)}^{2}\chi^{2}\left({\bar{F}}_{\eta },{\bar{F}}_{0}\right)\\ & = & \mathrm{CI}(\eta ), \end{array}$$
which proves Part (i). Part (ii) can be proved in an analogous manner. □

### 5.2. Harmonic Mixture Distribution

The harmonic mixture (HM) distribution based on survival functions ${\bar{F}}_{1}$ and ${\bar{F}}_{2}$ is given by
(22)$$\begin{array}{c} {\bar{F}}_{\eta }\left(x\right)=\frac{{\bar{F}}_{2}\left(x\right){\bar{F}}_{1}\left(x\right)}{\eta {\bar{F}}_{2}\left(x\right)+\left(1-\eta \right){\bar{F}}_{1}\left(x\right)},\;\eta \in \left(0,1\right). \end{array}$$

For more details about harmonic mixture distributions, one may refer to Schmidt [21]. The CRQF information measure about parameter η in (Equation 22) is given by
(23)$$\begin{array}{ccc} {\mathrm{CI}}_{q}\left(\eta \right) & = & \int {\left(\frac{\partial \log_{q}{\bar{F}}_{\eta }\left(x\right)}{\partial \eta }\right)}^{2}{\bar{F}}_{\eta }\left(x\right)dx\\ & = & \int \frac{{\left({\bar{F}}_{2}\left(x\right)-{\bar{F}}_{1}\left(x\right)\right)}^{2}}{{\left(\eta {\bar{F}}_{2}+\left(1-\eta \right){\bar{F}}_{1}\left(x\right)\right)}^{1-2q}}{\left({\bar{F}}_{1}\left(x\right){\bar{F}}_{2}\left(x\right)\right)}^{2q+1}dx. \end{array}$$

**Theorem** **8.**
*An upper bound for the CRQF information measure in (Equation 23), for q≥0, is given by*

$$\begin{array}{c} {\mathrm{CI}}_{q}\left(\eta \right)\leq \frac{8}{2q+1}M_{\frac{1}{2}}\left(\chi_{2q}^{2}\left({\bar{F}}_{T},{\bar{F}}_{1}\right),\chi_{2q}^{2}\left({\bar{F}}_{T},{\bar{F}}_{2}\right)\right), \end{array}$$

*where*

$${\bar{F}}_{T}\left(x\right)=\frac{{\bar{F}}_{\eta }\left(x\right)}{{\bar{F}}_{1}\left(x\right){\bar{F}}_{2}\left(x\right)}=\eta {\bar{F}}_{2}\left(x\right)+\left(1-\eta \right){\bar{F}}_{1}\left(x\right)$$


*and $M_{\frac{1}{2}}\left(.,.\right)$ is as defined earlier in Theorem 6.*


**Proof.** Because ${\bar{F}}_{T}\left(x\right)=\eta {\bar{F}}_{2}\left(x\right)+\left(1-\eta \right){\bar{F}}_{1}\left(x\right)$, it is readily seen that
(24)$$\begin{array}{c} {\bar{F}}_{2}\left(x\right)-{\bar{F}}_{T}\left(x\right)=\left(1-\eta \right)\left({\bar{F}}_{2}\left(x\right)-{\bar{F}}_{1}\left(x\right)\right), \end{array}$$
$$\begin{array}{c} {\bar{F}}_{1}\left(x\right)-{\bar{F}}_{T}\left(x\right)=\eta \left({\bar{F}}_{1}\left(x\right)-{\bar{F}}_{2}\left(x\right)\right). \end{array}$$By using these and (Equation 23), we find
$$\begin{array}{c} {\mathrm{CI}}_{q}\left(\eta \right)\leq \frac{2}{{\left(1-\eta \right)}^{2}\left(1+2q\right)}\chi_{2q}^{2}\left({\bar{F}}_{T},{\bar{F}}_{2}\right), \end{array}$$
$$\begin{array}{c} {\mathrm{CI}}_{q}\left(\eta \right)\leq \frac{2}{\eta^{2}\left(1+2q\right)}\chi_{2q}^{2}\left({\bar{F}}_{T},{\bar{F}}_{1}\right). \end{array}$$Adding the above two inequalities, we readily get
$$\begin{array}{ccc} {\mathrm{CI}}_{q}\left(\eta \right) & \leq & \frac{2}{2q+1}{\left(\sqrt{\chi_{2q}^{2}\left({\bar{F}}_{T},{\bar{F}}_{1}\right)}+\sqrt{\chi_{2q}^{2}\left({\bar{F}}_{T},{\bar{F}}_{2}\right)}\right)}^{2}\\ & = & \frac{8}{2q+1}M_{\frac{1}{2}}\left(\chi_{2q}^{2}\left({\bar{F}}_{T},{\bar{F}}_{1}\right),\chi_{2q}^{2}\left({\bar{F}}_{T},{\bar{F}}_{2}\right)\right), \end{array}$$
as required. □

### 5.3. HM Distribution Having Optimal Information under CR-$\chi^{2}$ Divergence Measure

In this section, we discuss the optimal information property of the harmonic mixture survival function in (Equation 22). For this purpose, we consider the optimization problem for cumulative residual chi-square divergence under three types of constraints. For more details about optimal information properties of some mixture distributions (arithmetic, geometric, and α−mixture distributions), one may refer to Asadi et al. [22] and the references therein.

**Theorem** **9.**
*Let $\bar{F}$, ${\bar{F}}_{0}$, and ${\bar{F}}_{1}$ be three survival functions. Then, the solution to the information problem*

(25)
$$\begin{array}{c} \min_{\bar{F}}\chi^{2}\left(\bar{F}:{\bar{F}}_{0}\right)\;\;subjectto\chi^{2}\left(\bar{F}:{\bar{F}}_{1}\right)=\theta ,\;\int \bar{F}\left(x\right)dx=\mu , \end{array}$$


*is the HM distribution in (Equation 22) with mixing parameter $\eta =\frac{1}{1+\lambda_{0}}$ and $\lambda_{0}>0$ is the Lagrangian multiplier.*


**Proof.** We use the Lagrange multiplier technique to find the solution of the optimization problem stated in (Equation 25). Hence, we have
$$\begin{array}{c} L\left(\bar{F},\lambda_{0},\lambda_{1}\right)=\int \frac{{\left(\bar{F}\left(x\right)-{\bar{F}}_{0}\left(x\right)\right)}^{2}}{{\bar{F}}_{0}\left(x\right)}dx+\lambda_{0}\int \frac{{\left(\bar{F}\left(x\right)-{\bar{F}}_{1}\left(x\right)\right)}^{2}}{{\bar{F}}_{1}\left(x\right)}dx+\lambda_{1}\int \bar{F}\left(x\right)dx. \end{array}$$Now, differentiating with respect to $\bar{F}$, we obtain
(26)$$\begin{array}{c} \frac{\partial }{\partial \bar{F}}L\left(\bar{F},\lambda_{0},\lambda_{1}\right)=2\frac{\bar{F}\left(x\right)-{\bar{F}}_{0}}{{\bar{F}}_{0}}+2\lambda_{0}\frac{\bar{F}\left(x\right)-{\bar{F}}_{1}}{{\bar{F}}_{1}}+\lambda_{1}. \end{array}$$Setting (Equation 26) to zero, we get the optimal survival function as
$$\begin{array}{c} \bar{F}\left(x\right)=\frac{1}{\frac{\eta }{{\bar{F}}_{0}\left(x\right)}+\frac{1-\eta }{{\bar{F}}_{1}\left(x\right)}}, \end{array}$$
where $\eta =\frac{1}{1+\lambda_{0}}$. □

**Theorem** **10.**
*Let $\bar{F}$, ${\bar{F}}_{0}$, and ${\bar{F}}_{1}$ be three survival functions. Then, the solution to the information problem*

(27)
$$\begin{array}{c} \min_{\bar{F}}\left\{w\chi^{2}\left(\bar{F}:{\bar{F}}_{1}\right)+\left(1-w\right)\chi^{2}\left(\bar{F}:{\bar{F}}_{2}\right)\right\}\;subjectto\;\int \bar{F}\left(x\right)dx=\mu ,\;0\leq w\leq 1, \end{array}$$


*is the HM distribution in (Equation 22) with mixing parameter η=w.*


**Proof.** Making use of the Lagrangian multiplier technique, and proceeding in the same way as in the proof of Theorem Equation 25, the required result can be obtained. □

**Theorem** **11.**
*Let $\bar{F}$, ${\bar{F}}_{0}$, and ${\bar{F}}_{1}$ be three survival functions and $T\left(X\right)=\frac{\bar{F}\left(X\right)}{{\bar{F}}_{2}\left(x\right)}$. Then, the solution to the information problem*

(28)
$$\begin{array}{c} \min_{\bar{F}}\chi^{2}\left(\bar{F}:{\bar{F}}_{0}\right)\;\;subjecttoE_{f_{e}}\left(T\left(X\right)\right)=\theta ,\;\int \bar{F}\left(x\right)dx=\mu , \end{array}$$


*is HM model with mixing parameter $\eta =\frac{1}{1+\lambda_{0}}$ and $\lambda_{0}>0$ is the Lagrangian multiplier, where $f_{e}$ is the equilibrium distribution as defined in Theorem 4.*


**Proof.** Making use of the Lagrangian multiplier technique, and proceeding the same way as in the proof of Theorem Equation 25, the required result is obtained. □

## 6. Jensen-Cumulative Residual $\chi^{2}$ and Parametric Version of Jensen-Cumulative Residual Fisher Divergence Measures

In this section, we first introduce the Jensen-cumulative residual $\chi^{2}$ divergence measure and then propose a parametric version of the Jensen-cumulative residual Fisher information in (Equation 7). Next, we show that these two Jensen-type divergence measures are connected through arithmetic mixture distributions.

### 6.1. Jensen-Cumulative Residual $\chi^{2}$ Divergence Measure

**Definition** **5.**
*Consider the survival functions ${\bar{F}}_{1,\theta },\cdots ,{\bar{F}}_{n,\theta }$ and ${\bar{G}}_{\theta }$. Then, the Jensen-cumulative residual $\chi^{2}$ (JCR-$\chi^{2}$) information measure is defined as*

(29)
$$\begin{array}{ccc} {J\chi^{2}}_{\alpha }\left({\bar{F}}_{1,\theta },\cdots ,{\bar{F}}_{n,\theta };{\bar{G}}_{\theta }\right) & = & \sum_{i=1}^{n}\alpha_{i}\chi^{2}\left({\bar{F}}_{i,\theta },{\bar{G}}_{\theta }\right)-\chi^{2}\left(\sum_{i=1}^{n}\alpha_{i}{\bar{F}}_{i,\theta },{\bar{G}}_{\theta }\right), \end{array}$$

*where $\alpha_{1},\cdots ,\alpha_{n}$ are non-negative real values with $\sum_{i=1}^{n}\alpha_{i}=1$.*


**Theorem** **12.**
*The JCR-$\chi^{2}$ information measure defined in (Equation 29) is non-negative.*


**Proof.** From the definition of the CR-$\chi^{2}$ measure, we have
$$\begin{array}{ccc} \sum_{i=1}^{n}\alpha_{i}\chi^{2}\left({\bar{F}}_{i,\theta },\bar{G}\right) & = & \sum_{i=1}^{n}\alpha_{i}\int \frac{{\left({\bar{F}}_{i,\theta }\left(x\right)-{\bar{G}}_{\theta }\left(x\right)\right)}^{2}}{{\bar{F}}_{i,\theta }\left(x\right)}dx\\ & = & \int {\bar{G}}_{\theta }^{2}\left(x\right)\sum_{i=1}^{n}\frac{\alpha_{i}}{{\bar{F}}_{i,\theta }\left(x\right)}dx+\int \sum_{i=1}^{n}\alpha_{i}{\bar{F}}_{i,\theta }\left(x\right)dx-2\int {\bar{G}}_{\theta }\left(x\right)dx \end{array}$$
and
$$\begin{array}{ccc} \chi^{2}\left(\sum_{i=1}^{n}\alpha_{i}{\bar{F}}_{i,\theta },{\bar{G}}_{\theta }\right) & = & \int \frac{{\left(\sum_{i=1}^{n}\alpha_{i}{\bar{F}}_{i,\theta }\left(x\right)-{\bar{G}}_{\theta }\left(x\right)\right)}^{2}}{\sum_{i=1}^{n}\alpha_{i}{\bar{F}}_{i,\theta }\left(x\right)}dx\\ & = & \int \frac{{\bar{G}}_{\theta }^{2}\left(x\right)}{\sum_{i=1}^{n}\alpha_{i}{\bar{F}}_{i,\theta }\left(x\right)}dx+\int \sum_{i=1}^{n}\alpha_{i}{\bar{F}}_{i,\theta }\left(x\right)dx-2\int {\bar{G}}_{\theta }\left(x\right)dx. \end{array}$$Upon making use of the above results, from the definition of the JCR-$\chi^{2}$ measure in (Equation 29), we find
$$\begin{array}{ccc} {J\chi^{2}}_{\alpha }\left({\bar{F}}_{1,\theta },\cdots ,{\bar{F}}_{n,\theta };{\bar{G}}_{\theta }\right) & = & \sum_{i=1}^{n}\alpha_{i}\chi^{2}\left({\bar{F}}_{i,\theta },\bar{G}\right)-\chi^{2}\left(\sum_{i=1}^{n}\alpha_{i}{\bar{F}}_{i,\theta },{\bar{G}}_{\theta }\right)\\ & = & \sum_{i=1}^{n}\alpha_{i}\int \frac{{\left({\bar{F}}_{i,\theta }\left(x\right)-{\bar{G}}_{\theta }\left(x\right)\right)}^{2}}{{\bar{F}}_{i,\theta }\left(x\right)}dx-\int \frac{{\left(\sum_{i=1}^{n}\alpha_{i}{\bar{F}}_{i,\theta }\left(x\right)-{\bar{G}}_{\theta }\left(x\right)\right)}^{2}}{\sum_{i=1}^{n}\alpha_{i}{\bar{F}}_{i,\theta }\left(x\right)}dx\\ & = & \int {\bar{G}}_{\theta }^{2}\left(x\right)\sum_{i=1}^{n}\frac{\alpha_{i}}{{\bar{F}}_{i,\theta }\left(x\right)}dx-\int \frac{{\bar{G}}_{\theta }^{2}\left(x\right)}{\sum_{i=1}^{n}\alpha_{i}{\bar{F}}_{i,\theta }\left(x\right)}dx\\ & = & \int {\bar{G}}_{\theta }^{2}\left(x\right)\left(\sum_{i=1}^{n}\frac{\alpha_{i}}{{\bar{F}}_{i,\theta }\left(x\right)}-\frac{1}{\sum_{i=1}^{n}\alpha_{i}{\bar{F}}_{i,\theta }\left(x\right)}\right)dx. \end{array}$$Finally, from the arithmetic mean-harmonic mean inequality (see Theorem 5.1 of Cvetkovski [23]), we get
$$\begin{array}{ccc} {J\chi^{2}}_{\alpha }\left({\bar{F}}_{1,\theta },\cdots ,{\bar{F}}_{n,\theta };{\bar{G}}_{\theta }\right) & = & \int {\bar{G}}_{\theta }^{2}\left(x\right)\left(\sum_{i=1}^{n}\frac{\alpha_{i}}{{\bar{F}}_{i,\theta }\left(x\right)}-\frac{1}{\sum_{i=1}^{n}\alpha_{i}{\bar{F}}_{i,\theta }\left(x\right)}\right)dx\geq 0, \end{array}$$
as required. □

### 6.2. Parametric Version of Jensen-Cumulative Residual Fisher Information Divergence

In this subsection, we introduce a parametric form of the JCRQF information measure in (Equation 7).

**Definition** **6.**
*Consider the survival functions ${\bar{F}}_{1,\theta },\cdots ,{\bar{F}}_{n,\theta }$. Then, a parametric form of JCRQF (P-JCRF) information measure about parameter θ, for non-negative real values $\alpha_{1},\cdots ,\alpha_{n}$ with $\sum_{i=1}^{n}\alpha_{i}=1$, is defined as*

(30)
$$\begin{array}{ccc} \mathrm{JCI}({\bar{F}}_{1,\theta },\cdots ,{\bar{F}}_{n,\theta },\alpha ) & = & \sum_{i=1}^{n}\alpha_{i}{\mathrm{CI}}_{\theta }\left({\bar{F}}_{i,\theta }\right)-{\mathrm{CI}}_{\theta }\left(\sum_{i=1}^{n}\alpha_{i}{\bar{F}}_{i,\theta }\right), \end{array}$$

*where*

$$\begin{array}{ccc} {\mathrm{CI}}_{\theta }\left({\bar{F}}_{i,\theta }\right) & = & \int {\left(\frac{\partial \log{\bar{F}}_{i,\theta }\left(x\right)}{\partial \theta }\right)}^{2}{\bar{F}}_{i,\theta }\left(x\right)dx,\;i=1,\cdots ,n. \end{array}$$



**Theorem** **13.**
*The P-JCRF information measure in (Equation 30) can be represented as a mixture of CD measures in (Equation 6) as*

$$\begin{array}{c} \mathrm{JCI}\left({\bar{F}}_{1,\theta },\cdots ,{\bar{F}}_{n,\theta },\alpha \right)=\sum_{i=1}^{n}\alpha_{i}\mathrm{CD}\left({\bar{F}}_{i,\theta },{\bar{F}}_{T,\theta }\right), \end{array}$$

*where ${\bar{F}}_{T,\theta }=\sum_{i=1}^{n}\alpha_{i}{\bar{F}}_{i,\theta }$ is the weighted survival function.*


**Proof.** From the definition in (Equation 30), we have
$$\begin{array}{ccc} \mathrm{JCI}({\bar{F}}_{1,\theta },\cdots ,{\bar{F}}_{n,\theta },\alpha ) & = & \sum_{i=1}^{n}\alpha_{i}{\mathrm{CI}}_{\theta }\left({\bar{F}}_{i,\theta }\right)-{\mathrm{CI}}_{\theta }\left(\sum_{i=1}^{n}\alpha_{i}{\bar{F}}_{i,\theta }\right)\\ & = & \sum_{i=1}^{n}\alpha_{i}\int_{0}^{\infty }{\left\{\frac{\partial \log{\bar{F}}_{i,\theta }\left(x\right)}{\partial \theta }\right\}}^{2}{\bar{F}}_{i,\theta }\left(x\right)dx\\ & & -\int_{0}^{\infty }{\left\{\frac{\partial \log\sum_{i=1}^{n}\alpha_{i}{\bar{F}}_{i,\theta }\left(x\right)}{\partial \theta }\right\}}^{2}\sum_{i=1}^{n}\alpha_{i}{\bar{F}}_{i,\theta }\left(x\right)dx. \end{array}$$Furthermore, we have
$$\begin{array}{ccc} \sum_{i=1}^{n}\alpha_{i}\mathrm{CD}\left({\bar{F}}_{i,\theta },{\bar{F}}_{T,\theta }\right) & = & \sum_{i=1}^{n}\alpha_{i}\int_{0}^{\infty }{\left\{\frac{\partial \log{\bar{F}}_{i,\theta }\left(x\right)}{\partial \theta }-\frac{\partial \log\sum_{i=1}^{n}\alpha_{i}{\bar{F}}_{i,\theta }\left(x\right)}{\partial \theta }\right\}}^{2}{\bar{F}}_{i,\theta }\left(x\right)dx\\ & = & \sum_{i=1}^{n}\alpha_{i}\int_{0}^{\infty }\{{\left(\frac{\partial \log{\bar{F}}_{i,\theta }\left(x\right)}{\partial \theta }\right)}^{2}-2\left(\frac{\partial \log{\bar{F}}_{i,\theta }\left(x\right)}{\partial \theta }\right)\frac{\partial \log\sum_{i=1}^{n}\alpha_{i}{\bar{F}}_{i,\theta }\left(x\right)}{\partial \theta }\\ & & +{\left(\frac{\partial \log\sum_{i=1}^{n}\alpha_{i}{\bar{F}}_{i,\theta }\left(x\right)}{\partial \theta }\right)}^{2}\}{\bar{F}}_{i,\theta }\left(x\right)dx\\ & = & \sum_{i=1}^{n}\alpha_{i}\int_{0}^{\infty }{\left\{\frac{\partial \log{\bar{F}}_{i,\theta }\left(x\right)}{\partial \theta }\right\}}^{2}{\bar{F}}_{i,\theta }\left(x\right)dx\\ & & -\int_{0}^{\infty }{\left\{\frac{\partial \log\sum_{i=1}^{n}\alpha_{i}{\bar{F}}_{i,\theta }\left(x\right)}{\partial \theta }\right\}}^{2}\sum_{i=1}^{n}\alpha_{i}{\bar{F}}_{i,\theta }\left(x\right)dx, \end{array}$$
as required. □

### 6.3. Connection between the P-JCRF Information and JCR-$\chi^{2}$ Divergence Measures

Let ${\bar{F}}_{1},\cdots ,{\bar{F}}_{n}$ and $\bar{G}$ be arbitrary continuous survival functions. Consider the arithmetic mixture distributions with survival functions
(31)$$\begin{array}{c} {\bar{H}}_{i,\Lambda }\left(x\right)=\Lambda {\bar{F}}_{i}\left(x\right)+\left(1-\Lambda \right)\bar{G}\left(x\right),\;i=1,\cdots ,n, \end{array}$$
and
(32)$$\begin{array}{c} {\bar{H}}_{\alpha ,\Lambda }\left(x\right)=\sum_{i=1}^{n}\alpha_{i}{\bar{H}}_{i,\Lambda }\left(x\right)=\Lambda \sum_{i=1}^{n}\alpha_{i}{\bar{F}}_{i}\left(x\right)+\left(1-\Lambda \right)\bar{G}\left(x\right), \end{array}$$
where $\alpha_{1},\cdots ,\alpha_{n}$ are non-negative real values with $\sum_{i=1}^{n}\alpha_{i}=1$ and 0≤Λ≤1.

The JCRQF information measure of survival functions ${\bar{H}}_{1,\Lambda },\cdots ,{\bar{H}}_{1,\Lambda }$, about the mixing parameter Λ, for 0≤Λ≤1, is given by
(33)$$\begin{array}{ccc} \mathrm{JCI}({\bar{H}}_{1,\Lambda },\cdots ,{\bar{H}}_{n,\Lambda };\alpha ) & = & \sum_{i=1}^{n}\alpha_{i}{\mathrm{CI}}_{\Lambda }\left({\bar{H}}_{i,\Lambda }\right)-{\mathrm{CI}}_{\Lambda }\left(\sum_{i=1}^{n}\alpha_{i}{\bar{H}}_{i,\Lambda }\right). \end{array}$$

**Theorem** **14.**
*The connection between ${J\chi^{2}}_{\alpha }\left({\bar{F}}_{1,\theta },\cdots ,{\bar{F}}_{n,\theta };{\bar{G}}_{\theta }\right)$ in (Equation 29) and $\mathrm{JCI}({\bar{H}}_{1,\Lambda },\cdots ,{\bar{H}}_{n,\Lambda };\alpha )$ in (Equation 33) is given by*

$$\begin{array}{ccc} \mathrm{JCI}({\bar{H}}_{1,\Lambda },\cdots ,{\bar{H}}_{n,\Lambda };\alpha ) & = & \frac{1}{\Lambda^{2}}{J\chi^{2}}_{\alpha }\left({\bar{H}}_{1,\Lambda },\cdots ,{\bar{H}}_{n,\Lambda };\bar{G}\right). \end{array}$$



**Proof.** From the definition of $\mathrm{JCI}({\bar{H}}_{1,\Lambda },\cdots ,{\bar{H}}_{n,\Lambda };\alpha )$ in (Equation 33), we find$$\begin{array}{ccc} \mathrm{JCI}({\bar{H}}_{1,\Lambda },\cdots ,{\bar{H}}_{n,\Lambda };\alpha ) & = & \sum_{i=1}^{n}\alpha_{i}{\mathrm{CI}}_{\Lambda }\left({\bar{H}}_{i,\Lambda }\right)-{\mathrm{CI}}_{\Lambda }\left(\sum_{i=1}^{n}\alpha_{i}{\bar{H}}_{i,\Lambda }\right)\\ & = & \sum_{i=1}^{n}\alpha_{i}\int {\left(\frac{\partial \log{\bar{H}}_{i,\Lambda }\left(x\right)}{\partial \Lambda }\right)}^{2}{\bar{H}}_{i,\Lambda }\left(x\right)dx-\int {\left(\frac{\partial \log{\bar{H}}_{i,\Lambda }\left(x\right)}{\partial \Lambda }\right)}^{2}{\bar{H}}_{i,\Lambda }\left(x\right)\left(x\right)dx\\ & = & \sum_{i=1}^{n}\alpha_{i}\int {\left(\frac{\partial \log\left\{\Lambda {\bar{F}}_{i}\left(x\right)+\left(1-\Lambda \right)\bar{G}\left(x\right)\right\}}{\partial \Lambda }\right)}^{2}{\bar{H}}_{i,\Lambda }\left(x\right)dx\\ & & -\int {\left(\frac{\partial \log\left\{\sum_{i=1}^{n}\alpha_{i}{\bar{H}}_{i,\Lambda }\left(x\right)\right\}}{\partial \Lambda }\right)}^{2}{\bar{H}}_{\alpha ,\Lambda }\left(x\right)dx\\ & = & \frac{1}{\Lambda^{2}}\sum_{i=1}^{n}\alpha_{i}\int \frac{{\left({\bar{H}}_{i,\Lambda }\left(x\right)-\bar{G}\left(x\right)\right)}^{2}}{{\bar{H}}_{i,\Lambda }\left(x\right)}dx-\frac{1}{\Lambda^{2}}\int \frac{{\left({\bar{H}}_{\alpha ,\Lambda }\left(x\right)-\bar{G}\left(x\right)\right)}^{2}}{{\bar{H}}_{\alpha ,\Lambda }\left(x\right)}dx\\ & = & \frac{1}{\Lambda^{2}}\left\{\sum_{i=1}^{n}\alpha_{i}\chi^{2}\left({\bar{H}}_{i,\Lambda },\bar{G}\right)-\chi^{2}\left({\bar{H}}_{\alpha ,\Lambda },\bar{G}\right)\right\}\\ & = & \frac{1}{\Lambda^{2}}{J\chi^{2}}_{\alpha }\left({\bar{H}}_{1,\Lambda },\cdots ,{\bar{H}}_{n,\Lambda };\bar{G}\right), \end{array}$$
as required. □

## 7. Application of CRQF Information Measure

We now demonstrate an application of the CRQF information measure to image processing. Let $X_{1},\cdots ,X_{n}$ be a random sample from density *f* with corresponding CDF *F*. The kernel estimate of density *f*, based on kernel function *K* with bandwidth h>0, is given by
(34)$$\begin{array}{c} \hat{f}\left(x\right)=\frac{1}{nh}\sum_{i=1}^{n}K\left(\frac{x-X_{i}}{h}\right). \end{array}$$

Further, the non-parametric estimate of the survival function $\bar{F}\left(x\right)$, at a given point *x*, is given by
(35)$$\begin{array}{c} \hat{\bar{F}}\left(x\right)=\frac{1}{n}\sum_{i=1}^{n}I\left(X_{i}>x\right), \end{array}$$
where *I* is the indicator function taking the value 1 if the condition inside brackets is satisfied and 0 otherwise. Then, the integrated non-parametric estimate of ${\mathrm{CI}}_{q}\left(\bar{F}\right)$ in (Equation 11) is given by
(36)$$\begin{array}{c} \hat{{\mathrm{CI}}_{q}}\left(\bar{F}\right)=\int \frac{{\hat{f}}^{2}\left(x\right)}{{\left[\hat{\bar{F}}\left(x\right)\right]}^{1-2q}}dx=\frac{1}{h^{2}n^{1+2q}}\int \frac{{\left(\sum_{i=1}^{n}K\left(\frac{x-X_{i}}{h}\right)\right)}^{2}}{{\left(\sum_{i=1}^{n}I\left(X_{i}>x\right)\right)}^{1-2q}}dx. \end{array}$$

From (Equation 36) and with the use of Gaussian kernel $K\left(u\right)=\frac{1}{\sqrt{2\pi }}e^{-\frac{u^{2}}{2}}$, we have
(37)$$\begin{array}{c} \hat{{\mathrm{CI}}_{q}}\left(\bar{F}\right)=\frac{1}{2\pi h^{2}n^{1+2q}}\int \frac{{\left(\sum_{i=1}^{n}e^{\frac{-1}{2}{\left(\frac{x-X_{i}}{h}\right)}^{2}}\right)}^{2}}{{\left(\sum_{i=1}^{n}I\left(X_{i}>x\right)\right)}^{1-2q}}dx. \end{array}$$

Thus, from (Equation 37) and using the Cavalieri-Simpson rule for numerical integration, the empirical estimate of the CRQF information measure can be obtained.

Next, we provide an example of image processing and compute the CRQF information and Fisher information (FI) measures for the original picture and its adjusted versions. Figure 2 shows a sample picture of two parrots (original picture) labeled as *X* and three adjusted versions of the original picture labeled as *Y* (increasing brightness), *Z* (increasing contrast), and *W* (gamma corrected). The available data of the main picture are 768×512
cells and the gray level of each cell has a value between 0 (black) and 1 (white). In order to examine the amount of content CRQF information measure of the original picture and compare with the information values of adjusted versions, we consider three cases as *Y*(=X+0.3), *Z*(=2X), and *W*($=\sqrt{X}$). For pertinent details, see EBImage package in R software (Pau et al. [24]).

We have plotted in Figure 3 the extracted histograms with the corresponding empirical densities for pictures *X*, *Y*, *Z*, and *W*. As we can see from Figure 2 and Figure 3, the highest degree of similarity is first related to *W* and then to *Y*, whereas *Z* has the highest degree of divergence with respect to the original picture *X*. We have presented the CRQF information (for selected values of q=0.55 and 0.7) and Fisher information (FI) measures for all four pictures in Table 1. It is easily seen that both information measures get increased when the similarity is decreased with respect to the original picture. This fact coincides with the minimum Fisher information principle. Therefore, the CRQF information measure can be considered as an efficient criteria, just as the Fisher information measure, in analyzing interior properties of the complex systems.

## 8. Concluding Remarks

In this paper, we have proposed cumulative residual *q*-Fisher (CRQF) information, *q*-hazard rate function (QHR), cumulative residual generalized $\chi^{2}$ (CRG-$\chi^{2}$) divergence, and Jensen-cumulative residual $\chi^{2}$ (JCR-$\chi^{2}$) divergence measures. We have shown that the CRQF information measure can be expressed in terms of expectation involving the *q*-hazard rate function. Further, we have established some results concerning the CRQF information and CRG-$\chi^{2}$ divergence measures. We have specifically shown that the first derivative of CRG-$\chi^{2}$ divergence with respect to the associated parameter is connected to cumulative residual entropy measure and, when its parameter tends to zero, it is connected to the variance of the ratio of two survival functions. We have also presented some results associated with the CRQF information measure for two well-known mixture models, namely, arithmetic mixture (AM) and harmonic mixture (HM) models. We have specifically shown that the CRQF information of AM and HM models can be expressed in terms of power mean of CRG-$\chi^{2}$ divergence measures. Interestingly, we have shown that the harmonic mixture model possesses optimal information under three optimization problems associated with the cumulative residual $\chi^{2}$ divergence measure. We have also proposed a Jensen-cumulative residual $\chi^{2}$ divergence and a parametric version of the Jensen-cumulative residual Fisher (P-JCRF) information measures and have shown that they are connected. Finally, we have described an application of the CRQF information measure by considering an example in image processing. It will naturally be of great interest to study empirical versions of these measures and their potential applications to inferential problems. We are currently looking into this problem and hope to report the findings in a future paper.

## Figures and Tables

**Figure 1 entropy-24-00341-f001:**
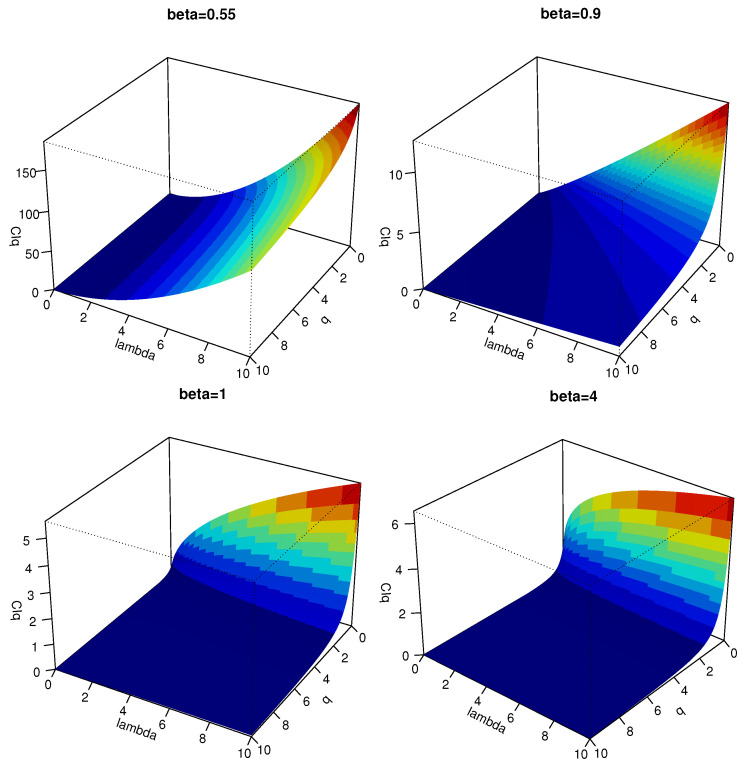
3D plots of the CRQF information measure in (Equation 12) for some selected choices of the Weibull parameters.

**Figure 2 entropy-24-00341-f002:**
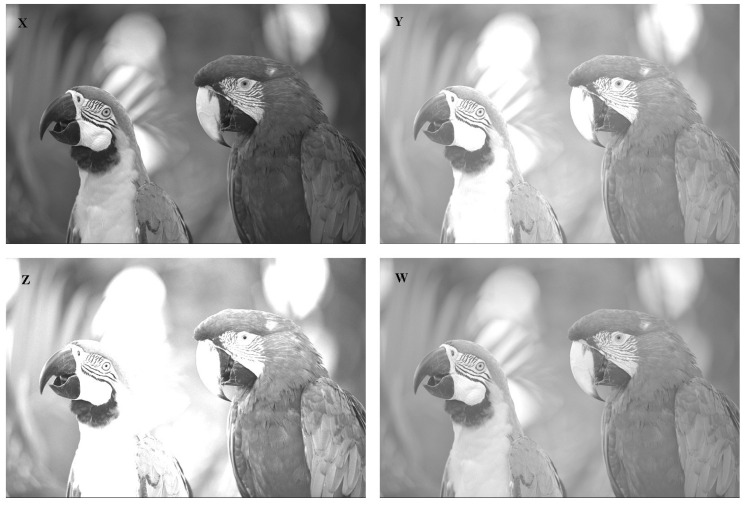
Sample picture of two parrots with its adjustments.

**Figure 3 entropy-24-00341-f003:**
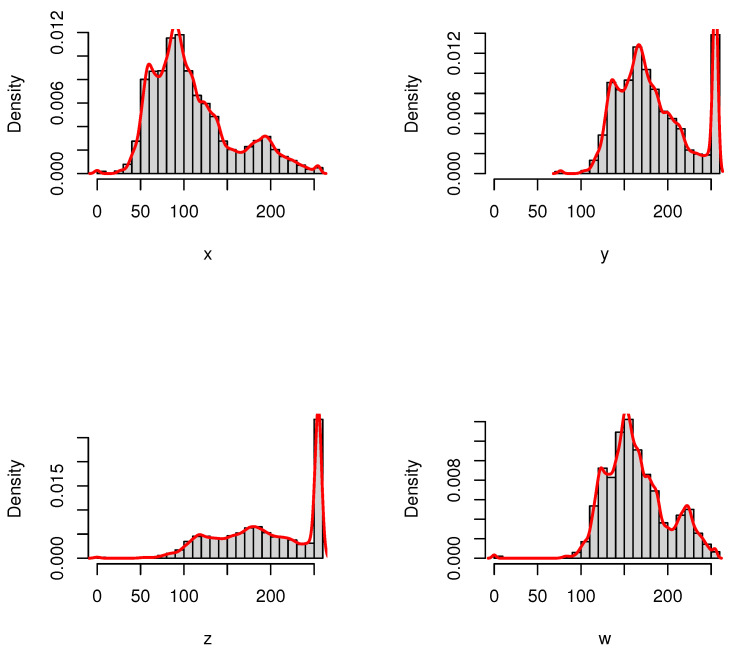
The histograms and the corresponding empirical densities for pictures *X*, *Y*, *Z*, and *W*.

**Table 1 entropy-24-00341-t001:** The CRQF information and FI measures.

	**CRQF (q=0.55)**	**CRQF (q=0.7)**	**FI**
X	0.0068	0.0057	0.0039
Y	0.0085	0.0066	0.0447
Z	0.0096	0.0071	0.0475
W	0.0082	0.0067	0.0052

## Data Availability

Data sharing not applicable.

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
