# Peer review of "Cumulative Residual q-Fisher Information and Jensen-Cumulative Residual χ2 Divergence Measures"

_entropy, 2022, doi:10.3390/e24030341_

Round 1
Reviewer 1 Report
In this work, the authors define cumulative residual q-Fisher (CRQF) information measures for model parameter as well as for the survival function of the underlying random variables. They also propose q-hazard rate (QHR) function via q-logarithmic function as a new extension of hazard rate function. Then, the generalized cumulative residual χ2 divergence measure between two survival functions is defined. Finally, they define Jensen-cumulative residual χ2 (JCR-χ2) measure and a
parametric version of the Jensen-cumulative residual Fisher information measure and then study their properties and inter-connections.
The paper is very well-written. Results and conclusions are properly described. I recommend the paper for publication in its present form.
Author Response
Thank you so much for your positive feedback, and we highly appreciate it.
Reviewer 2 Report
Comments
After carefully reading the proposed paper, this paper contains an interesting proposal; my overall impression is that the manuscript presents some results that could be useful in practice. I recommend the publication of this paper after some major comments:
My comments are:
- More details in the proof of Theorem 3.2 and Theorem 4.1 must be added.
- More applications must be added in the paper, there is no theoretical research only at this time. Recent data should be used and the proposed method should be compared with previous methods.
- More figures should be added in the paper to explain and illustrate the paper applications.
- In this paper, several concepts have been mentioned such as Cumulative residual q-Fisher information measure, q-Hazard rate function, generalized cumulative residual χ2 divergence measures, arithmetic mixture model, harmonic mixture model, Jensen-cumulative residual χ2 and parametric version of Jensen-cumulative residual Fisher divergence measures. Please add some numerical examples.
- Check for typographical errors.

Author Response
First of all, we thank the reviewers for their valuable comments on the earlier version of this manuscript, which led to an improvement in the manuscript entitled "Cumulative residual $q$-Fisher information and Jensen-cumulative residual $\chi^2$ divergence measures". According to reviewers' comments, we have prepared a new version of this article. Here is the point-by-point response to reviewers' comments.

Round 2
Reviewer 2 Report
Thanks a lot for the good responses to my comments.